# Vulnerability of bridges to scour: insights from an international expert elicitation workshop

Rob Lamb[1,2], Willy Aspinall[3,4], Henry Odbert[4], Thorsten Wagener[5,6]

[1]JBA Trust, South Barn, Skipton, BD23 3AE, UK
5  [2]Lancaster Environment Centre, Lancaster University, LA1 4YQ, UK
[3]Aspinall and Associates, UK
[4]School of Earth Sciences, University of Bristol, Bristol, UK
[5]Civil Engineering, University of Bristol, UK
[6]Cabot Institute, University of Bristol, UK

*Correspondence to*: Rob Lamb (rob.lamb@jbatrust.org)

**Abstract.** Scour (localised erosion) during flood events is one of the most important threats to bridges over rivers and estuaries, and has been the cause of numerous bridge failures, with damaging consequences. Mitigation of the risk of bridges being damaged by scour is therefore important for many infrastructure owners, and is supported by industry guidance. Even after 15  mitigation, some residual risk remains, though its extent is difficult to quantify because of the uncertainties inherent in the prediction of scour and the assessment of the scour risk. This paper summarises findings of an international expert workshop on bridge scour risk assessment exploring uncertainties about the vulnerability of bridges to scour. Two specialised structured elicitation methods were applied to explore the factors that experts in the field consider important in assessing scour risk, and to derive pooled expert judgements of bridge failure probabilities conditional on a range of assumed scenarios describing flood 20  event severity, bridge and watercourse types and risk mitigation protocols. The experts' judgements broadly align with industry good practice, but indicate significant uncertainty about quantitative estimates of bridge failure probabilities, reflecting the difficulty in assessing the residual risk of failure. The data and findings presented here could provide useful context for the development of generic scour fragility models, and their associated uncertainties.

## 1. Introduction

25  This paper summarises the outcomes of an international expert elicitation workshop on bridge scour risk assessment held in London in 2015. The workshop brought together 19 experts from organisations in the UK (12 experts), USA (5), New Zealand (1) and Canada (1), including representatives from industry (9 experts), academic researchers (5), and public agencies (5). Our ambition was to explore, in quantitative terms, uncertainties about the vulnerability of bridges to scour, with the ultimate aim being to inform the development of fragility functions that may be applied within a broad scale risk modelling framework 30  (where "broad scale" indicates modelling over an extensive network of assets rather than detailed, site-specific risk assessment, for example see Hall et al., 2016).

Scour refers to localised erosion that can undermine the foundations of bridges where they cross water. It is associated with high flows around the bridge piers, abutments and surrounding channel reaches, especially during flood events. The loss of support and consequent foundation movement caused by scour can result in costly damage to the structure, service restrictions, 35  and perhaps more importantly compromised safety for users of a bridge. In extreme cases the bridge structure may collapse. A critical threat to infrastructure around the world, scour is cited as the most common cause of bridge failure (Kirby et al., 2015); its importance is discussed further in Sect. 2 below.

Scour risk is managed through the application of assessment, monitoring and maintenance protocols, which are reviewed in Sect. 2. These protocols are undoubtedly effective in reducing risk by prioritising scour protection works, helping to spot 40  incipient problems and triggering maintenance or other mitigation actions when needed. Even so, the evidence of occasional

scour-related bridge failures indicates that some residual risk remains. This residual risk is difficult to manage, representing as it does a combination of rare events and uncertainties about the actual (as opposed to designed) response of assets to flooding. A generic framework for assessing the risk in terms of uncertain failure probabilities is outlined in Sect. 3.

The combination of infrequent natural drivers in the form of flood events, complex physical processes, and the difficulties, costs and uncertainties associated with measurements mean that it is difficult to quantify scour risk with confidence and, in particular, to extrapolate from historical or experimental evidence to more extreme situations. In these circumstances, the knowledge and judgement of experts constitutes an especially valuable source of information that can be harnessed to augment data from other sources. A formal process of elicitation was applied to develop a synthesis of current knowledge from expert judgements.

The elicitation methodology, described in Sect. 4, was a two-stage process. In the first stage, a categorical approach was used to examine which factors determine the likelihood of scour at a bridge, and how experts think those factors should be ranked in importance. The second stage involved a quantitative assessment of bridge failure probabilities for a range of plausible scenarios under stated conditions and assumptions. The elicitation techniques included methods to weight information from the group of experts so as to promote the most accurate and unbiased judgement of uncertainty, using control questions to 'calibrate', jointly, the statistical accuracy and informativeness of the experts' uncertainty judgements. These are traits that can differ – sometimes substantially – from one expert to another, and can be adjusted for by empirical scoring rules to generate an optimal group decision. Results of the elicitation are presented in Sect. 5 and discussed in Sect. 6, highlighting both implications for scour risk management and methodological conclusions relating to the process of expert elicitation.

## 2. Motivation

Scour is well-known to be an important hazard. A survey of notable bridge failures around the world by Smith (1976) found that almost half were associated with "flood and foundation movement", including collapses at many different types of bridges. In the US, scour is thought to be the most common cause of highway bridge failures (Kattell and Eriksson, 1998, Johnson, 1999). Using the US National Bridge Inventory, Cook (2014) also found the most likely cause of bridge collapses to be "hydraulic in nature", mostly scour, and determined that collapses caused by hydraulic factors were not related to the age of the bridge.

In the UK, on the rail network alone, more than 100 bridge collapses since 1843 have been attributed to scour in rivers and estuaries, causing 15 fatalities (Rail Safety and Standards Board, 2005, van Leuwen and Lamb, 2014). Recent cases include the collapse at Glanrhyd, Wales, in 1987, which led to the deaths of four people when part of a passenger train fell into the River Towy, and the failure of the Lower Ashenbottom viaduct in Lancashire, in June 2002. During the 2009 floods in Cumbria, UK, seven road and foot bridges failed due to a combination of scour and hydrodynamic loading, with the collapse of the Northside road bridge in Workington causing one fatality and significant disruption to communities. More recently 131 bridges were damaged during flooding in the same region, many because of scour (Cumbia County Council, 2016, Zurich Insurance Group and JBA Trust, 2016).

For UK rail bridges, the known bridge failures evince issues and uncertainties associated with assessment of scour risk, for example suggesting that some historical failures occurred after relatively minor flood events rather than extreme floods, perhaps because, prior to the introduction of modern scour management practices, there is more likely to have been undetected scour damage during a sequence of events that ultimately led to failure. Some uncertainties relate to data errors or missing information. However, the complexity of physical scour processes also leads to uncertainty in scour models. This complexity includes some inherently unpredictable factors such as the occurrence and severity of flood flows, and the accumulation of debris, which can amplify scour through additional turbulence and enhanced local flow velocities.

## 2.1. Risk-based management and industry practice

In the UK and many other countries, bridges are designed, inspected and maintained so as to withstand damage during events that are "reasonably foreseeable" over their intended service life (UK Roads Liaison Group, 2009). As with many infrastructure assets, there is a balance to be struck between the costs of reducing the risk of scour, likely damage, and expectations of public safety. Design guides, monitoring, inspections and detailed modelling all help to establish the level of resources needed to achieve an appropriate balance, noting that the question of what is "appropriate" is ultimately a matter of judgement and policy. Risk-based asset management concepts are widely applied to help inform these judgements. A risk assessment involves considering the outcomes that could result from a combination of drivers, such as extreme weather events, and the performance of assets when subjected to those events (Johnson et al., 2015). Kirby et al. (2015) and Arneson et al. (2012) give comprehensive guidance for scour risk management, including references to numerous industry and government agency scour management protocols, including the UK Design Manual for Roads and Bridges (Highways England, 2016), US National Bridge Inspection Standards (FHWA, 2016), and US Forest Service scour assessment process (Kattell and Eriksson, 1998).

Scour risk management guidance typically deals with uncertainty through a combination of quantitative and qualitative analysis within a tiered structure, where relatively inexpensive, rapid "high level" screening is used to prioritise further investment of resources for more detailed assessments at bridges where scour may be more likely to occur, or where its consequences may be worst. Multiple factors are typically considered at each level within a tiered assessment, including physical characteristics of the bridge structures, the watercourses that they cross, their wider flow and sediment regimes and historical observations or recent changes relating to scour.

Most guidance involves some probabilistic analysis, which is usually introduced through the estimation of potential scour depth for an assumed "design flood", specified in terms of an annual exceedance probability (AEP, or equivalently a return period, which, when expressed in units of years, is numerically equal to 1/AEP). The design flood scour estimate can be compared with an estimated or known foundation depth to calculate a risk score. Recommended design flood conditions for UK railway and road bridge scour assessments are 1/200 AEP. In the USA the design condition is typically 1/100 AEP, but with a margin of safety that the structure should not fail in a 1/500 AEP event (Kirby et al., 2015). The probabilistic analysis is one part of the broader scour risk assessment protocols set out in industry guidance.

In this study the objective is to focus on uncertainties and their role in the probabilistic analysis of scour. In contrast to a design event analysis, we seek to explore how uncertainty about scour risk could be captured through a generic fragility model for bridge failure probability, reflecting a range of loading conditions, and including possible increases in vulnerability following exposure to flooding.

We ask explicitly how a general probabilistic failure model of this type could be formulated. The underlying motivation is an interest in generalising from detailed understanding of scour at specific bridges to consider the risks aggregated over a network or portfolio of assets, to support analysis either for a "generic bridge", or in a distributed, network-scale model of risk. The former case represents situations in which there may be inadequate information to carry out a detailed risk assessment. The latter is important in the context of strategic decisions about future planning, investment and operations for various infrastructure systems (e.g. Hall et al., 2016). This type of generalisation may not be appropriate for application to engineering decisions at individual assets, but is relevant as part of the higher-level risk screening that forms one tier in scour management approaches applied in practice.

## 3. Risk analysis framework

In the case of scour risk, the underlying hazard events are flood flows to which bridges and their foundations may be vulnerable. The drivers are uncertain because of the stochastic nature of flood events, meaning that it is not known for certain whether a flood of some given level of severity or extremeness will be encountered during the design life of the bridge, or indeed in any

specified period of time. Compounding this, it is not certain that an asset will perform as intended in response to any particular event or sequence of events, especially when conditions exceed design specifications. Indeed, for assets of unknown age and origin there may be no applicable specification for the design, although retrospective assessments and structural improvements can be, and often are, made.

To assess the risk associated with scour thus requires an understanding of the type of events that could plausibly occur and how an asset might respond to them. Although there could be many ways to do this, we argue that a powerful and general approach is, if possible, to treat the flood hazard and the asset performance in terms of probabilities, which allows the risk assessment to be framed ultimately in terms of a probability distribution of outcomes.

A high-level conceptual risk model for bridge failure from scour is outlined in Fig. 1, where the processes that create the flood
hazard are described in terms of the probability distribution of some relevant load variable, and the response of the bridge is described by a fragility function, representing the probability of a failure occurring conditional on an assumed load level. Figure 1 maps directly onto well-established, generic risk modelling frameworks, including the source – pathway – receptor concept widely used in environmental risk assessment (Defra, 2011), the loading and fragility concepts of reliability analysis (Ellingwood, 2008, USACE, 2010) and the hazard – vulnerability – loss concepts often applied in natural hazard risk
assessments for insurance (Mitchell-Wallace et al., 2017).

The scour risk can be expressed in generic terms via the distribution function F[$Y(L,S)$] of possible outcomes $Y$ when a bridge is subjected to some load representing the source of the scour hazard, where $L$ is a random variable describing the relevant loading condition(s) and $S$ is a state variable that is used to describe the uncertain response of a bridge under a given load (e.g. $S = 1$ if the bridge "fails" due to scour and $S = 0$ otherwise). The distribution function $G(l) = \Pr[S = 1 \mid L \leq l]$ is the probability
of failure conditional on a load event $L = l$. At this point no precise definition of loading condition or failure is offered. Failure could legitimately be defined as catastrophic collapse of the bridge, or in terms of a failure to continue providing some specified level of service (e.g. safe passage for traffic). The function $G(l)$ can be called a fragility function, or more generally, a model of vulnerability, and is central to this analysis.

Our aim is to help inform the development of scour risk models by investigating two general motivating questions:

1) What are the most important factors that should be considered in assessing scour risk to bridges?

2) What are the failure probabilities associated with a range of possible loading conditions, and how uncertain are they?

The former question is intended to help explore what variables could and should be chosen to describe the loading condition(s) relevant to scour risk assessment. For an asset-specific model there may be an obvious loading condition, such as flood water level at the bridge, together with detailed data or models to help predict the performance of the structure. In a more generic analysis, the definition of the relevant load condition is not necessarily clear because the factors that matter most may vary from asset to asset. Whilst this study does not progress to a full description of fragility functions, the results may help to inform
their development by informing the choice of relevant loading conditions and providing a pooled expert assessment of failure probabilities and associated uncertainties.

## 4. Expert elicitation methodology

The two motivating questions posed above could be tackled through empirical analysis or modelling of data for specific bridges. Deterministic models exist to predict the scour depths at structures for prescribed conditions including equilibrium
scour, (Melville, 1997) and time-varying scour (Melville and Chiew, 1999, Coleman et al., 2003). Most scour prediction models are based either on physical hydraulic formulae with coefficients calibrated from laboratory or field data (e.g. Ettema

et al, 1998, Sheppard et al, 2014, Ng et al, 2015), or may have a statistical basis (e.g. Hong et al., 2012). Models inferred from empirical observations inevitably carry some uncertainty (see Zevenbergen, 2010, for a comparison of differences between three established formulae in over 2500 scour calculations), which can be expected both to increase and to be difficult to quantify when generalising beyond the sample or type of structure used in the original inference.

In assessing the risk of scour failure over a broad network of assets and over an arbitrary time period, deterministic models for scour also need to be combined with analysis of the frequency or probability of hazardous flood events, (see, for example, Decò and Frangopol, 2011, Roca and Whitehouse, 2012), introducing further uncertainty inherent in the assessment of extremes. For a broad scale analysis some significant sources of uncertainty therefore remain that reflect the unpredictability of any given asset's actual performance under a range of conditions, and the generalisation from specific cases to generic
classes of structure for use in broader-scale risk analysis.

Inevitably, uncertainty has a major influence on a risk assessment and on any associated decisions in circumstances such as this where rare events are being considered. In these situations, there may be a need to appeal to the judgement and advice of experts, and some subjectivity is inevitable in the interpretation of terminology and data.

Soliciting expert advice for decision support is not new. Often it has been pursued on an informal basis. In this study, a
structured approach has been taken to elicit expert judgements from a range of opinions such that a rational consensus emerges about appropriate levels of uncertainty to be used in risk analysis. The formalised elicitation methodologies we adopted are designed to tie results into stated and transparent methodological rules, with the goal of treating expert judgements in the same way as other scientific data in a formal decision process. Various methods for assessing and combining expert uncertainty have been described in the literature. Until recently, the most familiar approach has been one that advocates a group decision-
conferencing framework for eliciting opinions, but other approaches now exist for carrying out this process more objectively. Two elicitation methods were selected for this study, corresponding to the two motivating questions:

1)   Expert judgement on choice of variables to describe the loading conditions in scour vulnerability analysis:

A specialised variant of the survey method of paired comparison was selected to assess judgements about the relative importance of factors that control vulnerability to scour. Initially, the method involves presenting a list of items and asking each expert to express a preference or importance ranking for every pairwise combination of the items. Then, a unique probabilistic inversion technique (see Cooke and Misiewicz, 2007 for a discussion of the mathematical basis) is used to reveal the overall preference ordering of the items, both for each expert and for the group, along with a
numerical assessment of the logical coherence of the responses in terms of 'circular triads' in the experts' responses (i.e. if item A is ranked above item B by an expert, and B is ranked above C, then C should not be ranked above A). The software tool UNIBALANCE (Macutkiewicz and Cooke, 2006) was used to process experts' preferences as individuals and as a group, to construct a formal probabilistic group representation of the alternative views expressed through the paired comparison elicitation.  The UNIBALANCE analysis output provides objective measures of
confidence about the extent to which the experts believe it is possible to discriminate between alternative factors.

2)   Failure probabilities associated with a range of possible loading conditions, and associated uncertainties:

For uncertainty quantification, a structured expert judgment procedure formulated by Cooke (1991), known as the
"Classical Model", was adopted in this study. This approach is supported by a software package called EXCALIBUR (Cooke and Solomatine, 1992), available at www.lighttwist.net/wp/excalibur. This is a quantitative elicitation method used to assess numerical estimates of uncertain parameters or variables, in this case scour failure probabilities conditional on various stated assumptions.

The unique feature of this approach is that distinct weights are given to individual experts, based on a statistical test of the expert's ability to judge uncertainties, determined empirically by performance metrics derived from control questions. The main steps in the procedure for applying the Classical Model in practice are:

- A group of experts is selected by a problem owner and a facilitator, and an elicitation protocol is developed; this comprises a set of multiple 'seed items' (i.e. the control) and a set of 'target questions', both drawn from within the experts' field of knowledge;

- The experts assess the set of 'seed item' quantities; experts are not expected to know the true values but should be able to capture most of them by defining informative credible ranges. Taking their responses to the set of seed items, the experts are treated as statistical hypotheses and are scored with respect to statistical likelihood ('calibration') and informativeness, using theory and procedures described by Cooke (1991);

- These scores are combined to form individual performance weights using scoring rules formulated such that experts receive maximal weight by, and only by, stating their true degrees of belief;

- The elicitation protocol includes a set of 'target item' questions; in principle, these could be subject to possible measurement or observation but, in the problem owner's case, for one reason or another they are not amenable to such an approach; the only feasible recourse is to seek expert judgements;

- Experts are elicited individually regarding their uncertainty judgements for these target items. A weighted linear combination of their responses is calculated for each question using EXCALIBUR to provide a pooled result (known as a synthetic 'decision maker'), conditioned on the performance-weighted scores.

The latter is the key feature of this method. When it comes to attempting to resolve differences in expert judgments, searching for harmony of views by negotiation or conciliation can leave participants discomfited by the outcomes. Extensive experience (see below for references to previous case studies) overwhelmingly confirms that experts grow to favour the Classical Model approach because its performance measures are objective and amenable to diagnostic examination. The 'reward' nature of weights is very important. An expert's influence on the pooled result should not appear haphazard, and he/she should be discouraged from attempting to game the system by attempting to tilt his/her assessments to achieve a desired outcome. Thus, it is necessary to impose a formal scoring rule constraint on the weighing scheme. This means an expert achieves maximal expected weight by, and only by, stating assessments in conformity with their true scientific or technical beliefs.

The Classical Model approach has been extensively used elsewhere in natural hazards risk assessments (e.g. Bamber and Aspinall, 2013, Aspinall and Cooke, 2013, Ioannou et al., 2017) and in many other uncertainty-related problem areas (a summary of case histories using the procedure was given by Cooke and Goossens, 2008). Aspinall et al. (2016) evaluated the method in detail in the context of a global mega-elicitation for the World Health Organization and Colson and Cooke (2017) reviewed its use in a meta-analysis of 78 case studies.

In similar vein to the present study, an elicitation using paired comparison probabilistic inversion jointly with uncertainties elicited with the Classical Model was reported by Tyshenko et al. (2011) for an elicitation for prion disease risk, but the combination of these methods has not to our knowledge previously been documented in the natural hazards or civil engineering literature.

The two broad motivating questions introduced in Sect. 3 were explored in detail, using the methods discussed above, through a set of more specific elicitation questions detailed in Table 1, with the results presented in the next section.

## 5. Results

### 5.1 Question (1): What are the most important factors that should be considered in assessing scour risk to bridges?

In the first stage of the elicitation, and following some discussion of issues and available information, the experts in the group were asked to complete a series of paired comparisons structured around the following question: "What are the most important factors that should be considered in assessing scour risk to bridges?". In each case the probabilistic inversion technique was used to calculate a group score and associated uncertainty.

The factors to be ranked were proposed by the project team and amended following an initial group discussion at the workshop (Table 2). The initial discussion raised concerns that the risk assessment priorities for piers and abutments may differ. The analysis was therefore carried out twice, treating scour at bridge piers and scour at abutments as separate issues. Results are shown in Table 3 for each of the potential assessment factors, and in Fig. 2, where the scores for scour risk to abutments and piers can be compared.

In the expert group's view, the most important factors in assessing vulnerability to scour (though with only weak discrimination between the factors) were as follows:

1) Scour history, i.e. whether or not scour has been a problem in the past
2) The morphological regime in the watercourse, including removal of sediments and morphological instability
3) Characteristics of bridge structure, including foundation type and depth, and the degree to which the flow is constricted at the bridge
4) The existence of inspection and scour assessment policy, and existence of prior scour protection
5) Watercourse characteristics or changes that may be unpredictable (e.g. debris accumulation) or cause progressive change in vulnerability (e.g. weir removal), but may be detectable in time to intervene during or between flood events
6) Uncertainty in knowledge about the foundations
7) Attributes of the bridge structure other than the foundations and constriction of the flow (e.g. bridge type, bridge span, construction date)
8) Recent flood history

Generally, factors ranked as important in determining the risk of abutment scour were also ranked as similarly important for scour at piers. The presence of an oblique approach flow was considered markedly more important for scour at piers than at abutments, although of less importance than other factors considered, in both cases.

### 5.1.1 Definition of loading conditions for fragility functions

Further discussion led to a refined set of factors that might be proposed to define relevant loading conditions for a scour fragility function. The experts were asked to rank this list in order of relevance. Overall, the ranking scores (Table 4) are quite compressed, ranging from 0.31 for the existence of a 'Scour assessment procedure', implying this was judged to be of relatively low importance amongst the list of factors proposed for determining bridge vulnerability, to 0.65 for 'Frequency and amount of debris', which was of greatest concern. The factors appear subjectively to separate into three clusters of differing importance, comprising two, three and five factors, respectively, labelled A, B and C in Table 4. The uncertainty about the rank order is broadly consistent for all factors.

Five factors appear to emerge as a preferred group from which the load variable in a fragility function might be defined. One is related to debris load. The others relate to hydraulic conditions during a flood event, including flood flow, flood flow return period, flow velocity and also duration of high flow.

Flood flow, velocity and flood return period may be intrinsically linked. However, the return period, or, alternatively, exceedance probability is a more abstract measure of a load event's intensity, albeit one that is open to interpretation with respect to the choice of methods applied to define a "flood event" and estimate its probability. In contrast with the physical

parameters more usually considered for asset-specific scour assessments, a probabilistic definition of loading such as "flood return period" may be viewed as a standardised measure of the load intensity defined on a common scale (e.g. the annual exceedance probability or return period in years) regardless of the actual physical scale of the system (e.g. channel width and depth, typical flow rates, or upstream catchment area). The results also suggest there is value in further investigation of the role of event durations within scour fragility analysis and the possibility that sequences or clusters of high flow events may also be important, although it may be more complicated to incorporate these temporal factors within a fragility function.

### 5.1.2 Potential changes in scour vulnerability

Finally, the expert group was asked to consider factors judged to be important in determining how the risk of failure may change under different circumstances. The factors discussed, and the group's ranking of them, are in Table 5. In this case, there is greater spread in factor rankings, suggesting that the expert group was clearer about discriminating between factors that could be used to determine how scour risk may change. Change in inspection regime was identified as the most important factor.

Climate change did not emerge as an important consideration in the ranking scores. Post-hoc discussion with some members of the expert panel showed that the factor labelled "Climate Change affects frequent extreme rainfall" was interpreted variously as meaning "the impacts of climate change on failure risk in the next few years" or "the impacts on risk in the long term". In either case, detailed feedback suggests that there may be important contextual differences in relation to this question. In the USA, a typical bridge design standard may be based on a 1/100 annual probability storm, but with an expectation of withstanding a more extreme storm of 1/500 annual probability. Hence even if climate change projections point to an increase in storm severity, the factor of safety allows for some confidence that the bridge scour risk is not unacceptably increased. This remark was made in the context of a typical service life of 75 years, with a re-evaluation of the required design being planned at that point, in effect allowing for a degree of planned adaptation. One of the US experts observed that the UK experts may not be able to assume a specified design standard for older bridges, especially if their foundation depths are not known precisely, and therefore may be more sensitive to the risk of increased flooding in a changing climate.

The discussion above brings out some ambiguities within the group's pooled responses owing to different assumptions made by participants from different countries about terminology and design standards.

### 5.2 Question (2): Quantitative elicitation of failure probabilities, with uncertainties

In the quantitative elicitation, the expert group was asked to estimate bridge failure probabilities, associated with scour caused by flooding under a range of conditions. In each case, the experts were asked for lower, central and upper values, corresponding to their judgements about the 5th, 50th and 95th percentiles of the range within which the true failure probability lies. The individual responses were pooled, with and without weighting, using the Classical Model (Cooke, 1991).

The failure probabilities were requested under various different conditions relating to: flood return period; type of watercourse; type of bridge foundations; type of monitoring/inspection and maintenance policy in force ("maintenance"). The definitions of generic types of watercourse, foundation and "maintenance" regime generated lengthy debate, primarily reflecting geographical differences in emphasis between the UK and North American experts. The following definitions were eventually adopted as a working compromise with the general assent of the group. The group agreed to have in mind physiographic and climatic conditions typical of the UK context, i.e. predominantly a humid temperate climate and a mixture of upland and lowland rivers, and to exclude more extreme (by UK standards) environments such as large continental scale rivers, Alpine rivers or rivers flowing in arid regions.

Two generic types of watercourse were specified: 1) Unmanaged watercourse – no channel or upstream measures specifically designed to reduce scour risk (such as active vegetation management to reduce risk of debris or promote sediment stability);

2) Managed watercourse – actively managed to control or reduce scour risk (or for other primary purposes also serving to reduce scour risk).

Two generic foundation types were specified: 1) Shallow foundations – a class including some historical masonry structures in the UK, particularly in lowland rivers, where foundations may be shallow pads or piles; 2) Deep/bedrock – a class that would include modern deep piles and also historical structures build directly onto solid bedrock, for example some UK bridges over upland rivers.

Three potential asset management regimes were specified, one of which relates to current practice: 1) None – a counterfactual assumption (at least for UK, North America and regions with rigorous engineering codes) of no investment of resources in monitoring, inspection or maintenance of scour protection maintenance works; 2) Routine – an investment of resources roughly similar to present-day good practice in the UK, US, Canada or New Zealand; 3) Premium – a counterfactual and significantly enhanced level of investment in inspection, monitoring and maintenance, featuring pro-active, highly precautionary investments in maintenance and scour protection.

After much discussion, the workshop group settled on a definition of "failure" as damage caused by the flood event to the structure, foundations or approaches, probably due to scour, sufficient to: cause a threat to safety; disrupt service and require repair action; cause collapse or would cause collapse if left unattended. (Note that this is a less restrictive definition of failure than one in which only a catastrophic collapse of the structure would be considered.)

**5.2.1 Guide to interpreting the results**

Results of the elicitation are plotted in Fig. 3-5. In each case, the bars represent the range of the 5th to 95th percentile estimates pooled from the expert group. The bold lines and symbols are the result of pooling the experts' estimates with weightings applied based on the performance of each individual in assessing uncertainty through the calibration questions. The lighter grey lines and symbols are the equivalent estimates, but this time combined with equal weight afforded to each expert. Results have been plotted on a logarithmic scale because in some cases the estimated probability ranges cover several orders of magnitude.

**5.2.2 Event failure probabilities (fragility estimates)**

The pooled estimates of failure probabilities (Fig. 3) tend, as expected, to increase as the intensity of the flood event increases. The failure probabilities also appear to decrease with improving maintenance regime.

Differences in the central estimates of failure probability with respect to flood event return period, maintenance assumption or watercourse/foundation type are generally rather smaller than the uncertainty ranges associated with the estimates. Note that the ranges are quantile estimates and not associated with any prescribed error distribution. Clearly the expert group's assessment of uncertainty is to place wide margins on any fragility estimate. Indeed, it would be surprising if this were not the case, given the nature of the problem as posed.

Although set against a wide range of uncertainty, the estimates of failure probability appear to increase systematically as flood event return period increases, and in line with expectations if comparing an obviously more resilient scenario (e.g. bridge with deep/bedrock foundations and "premium" maintenance) with a more vulnerable one (e.g. a bridge with shallow foundations and no maintenance).

Different assumptions about the foundation/watercourse type seem to cause large variation in the estimates of the upper uncertainty bounds under no maintenance or routine maintenance, particularly for the more extreme flood events (100-year and 500-year return period); for example, comparing top left and bottom left panels in Fig. 3, noting the logarithmic scale.

In comparison with an equally-weighted group estimate, the performance-weighted estimates display more constrained uncertainty. In particular, this is marked for the 100-year flood event results, where the application of weighting conditioned on the calibration questions results in a much lower pooled estimate of the upper quantile (95th percentile) on failure probability.

Other than for the managed, deep/bedrock case, this "calibration" of the upper failure probability bounds is not accompanied by a downward shift in the lower bounds. For the more extreme, 500-year return period flood, the weighting against performance on calibration questions makes little difference; this would suggest that although accounting for individual experts' skill in assessing uncertainty may help to refine group judgements about moderate failure probabilities, it does not constrain the very wide uncertainty in judgements about failure probability under very extreme flood conditions.

### 5.2.3 Annual failure probabilities

The experts were also asked to give ranges for their estimates of the annual probability of failure, again considering the three notional "maintenance" regimes and the four foundation and watercourse types.

The results (Fig. 4) follow expected patterns in that larger failure probabilities were estimated for the shallow foundation cases than for deep foundations, estimated failure probabilities were higher for an unmanaged watercourse than a managed watercourse, and estimated failure probabilities decrease as the assumed maintenance regime improves

The overall effect of applying performance weighting, based on calibration questions, has been to constrain the ranges of uncertainty without causing marked changes in the central estimates of failure probability for most cases. It is interesting to note that this performance-weighted modulation of elicited ranges is much more pronounced for the cases that describe inherently more resilient bridges (i.e. deep/bedrock foundations). An implication is that pooled estimates based on performance-weighted judgements appear to have resulted in a rather less precautionary judgement about uncertainty for the most resilient asset types.

Clearly the question, as it was posed, required the experts to make some general assumptions, either implicitly or explicitly, about the probability distribution of flood flows at a bridge and actual or inferred design standards. This lack of specific context to constrain those assumptions may account for some of the uncertainty expressed by the experts. A discussion was held about whether the annual failure probability is in fact determined completely by design standards (i.e. the as-built performance of the bridge matches the desired design standard perfectly), effectively removing uncertainty about bridge vulnerability. This view would appear to imply a standard of asset maintenance and that may be unachievable in practice and seems to be counter to the wide uncertainties about vulnerability to scour that emerged from the expert group elicitation. Empirically, historical evidence from the UK railway network shows that bridge failures have occurred under a wide range of flood conditions (van Leuwen and Lamb, 2014), suggesting that it is not appropriate to treat vulnerability deterministically.

### 5.2.4 Conditional event failure probabilities

The experts were asked to consider conditional failure probabilities for a generic bridge (defined below), when subjected to flood conditions of different levels of severity, conditional on the assumption that a preceding 100-year return period flood had already occurred, and with no intervening maintenance. The term "generic bridge" was taken to mean that variations in foundation, river characteristics or maintenance protocols were to be included as part of the uncertainty in the estimates. Pooled responses are shown in Fig. 5.

The pooled central estimates correlate with the severity of the flood event, as expected. For an extreme 1000-year event, the central estimate of the group is that there is more than a 50% chance of failure. However, the ranges express what is essentially a position of complete uncertainty about the most pessimistic (i.e. upper bound) judgement about the failure probability uncertainty, with the performance weighted group estimates differing little from the equally weighted estimates.

It can be seen that in the judgement of the group, the likelihood of a failure under extreme conditions of a sequence of 100-year flood followed by 1000-year flood is at least 1%. This is about 10,000 times more likely than the most optimistic pooled judgement made about failure probability for a minor, 5-year flood following after the 100-year event.

**5.2.5 Triggers for asset inspection**

As a supplementary question, experts were asked to make a judgement about a threshold flood return period that should trigger a new inspection. The pooled responses, shown in Table 6, indicate that the experts envisage a long upper tail in their judgement of uncertainty about a trigger threshold defined in this way. All experts express some belief within the elicited
uncertainty (5th to 95th percentile estimates) that an inspection trigger based on a probabilistic measure of flood severity could possibly be encountered with a probability of close to 1.0 in any given year (return period ≈ 1 year). When pooled with equal weights the group median response was to suggest an inspection threshold at a 26-year return period flood (1-in-26 annual exceedance probability), and that the inspection threshold might (at an upper, 95th percentile, limit of uncertainty) be set as high as a 318-year return period flood. This upper limit would indicate a considerably more relaxed inspection criterion than
scour assessment protocols in use today. However, when the pooled response is weighted according to the experts' judgement of uncertainties during the calibration exercise, the assessments become much more precautionary, with a median response that inspections be triggered by a flood of 5.6-year return period, with the 95th percentile estimate of the inspection trigger being a 48-year return period flood.

## 6. Discussion

The findings of the workshop are assessed below in four parts, relating to: the identification of factors considered important in determining the vulnerability of bridges to scour (Sect. 6.1); failure probabilities and associated uncertainties (Sect. 6.2); methodological considerations regarding the elicitation process (Sect. 6.3); and how the findings relate to current industry guidance on scour management (Sect. 6.4).

## 6.1 Choice of factors for scour vulnerability assessments

The findings of the workshop were well-aligned with current industry guidance on scour assessment, highlighting the importance of foundation depth, scour depth (either measured or predicted from modelling), river typology (i.e. whether a steep channel or lowland watercourse) and foundation material (e.g. clay, rock or of unknown type), which are all taken into consideration.

Additionally, the expert group identified other factors that are potentially important in assessing scour risk and that might be
given greater emphasis in risk assessment guidance. These factors highlight the potential influence of changes to a watercourse at and around a bridge: dredging or sand/gravel extraction; removal of weirs near bridge; and influence of flood defences.

The group also highlighted the importance of inspection and assessment regimes (i.e. the level of resources committed to scour monitoring and assessment, or changes in that commitment) in controlling the risk posed by bridge scour.

Risk factors relating to hydraulic conditions during flood events (flood flow magnitude, duration, and flow velocities around
the structure) and morphological regime (dredging) were consistently ranked by the group as important in determining scour vulnerability, although there was considerable ambiguity about the relative importance of many other factors, supporting the application of multi-factorial approaches to risk assessment.

In addition to variables expressed on physical scales, the return period (or exceedance probability) of a flood event was identified as a possible approach to define a generic loading condition for the development of bridge scour fragility functions.
Fragility functions are not incorporated into routine scour management guidance. The data presented here could be used to give some context to functions of this type should there be future work to develop reliability analysis models based on fragility concepts.

**6.2 Expert views on scour failure probabilities and associated uncertainties**

Experts' estimates of failure probability appear to increase systematically as the assumed loading, i.e. flood event severity, increases. Their failure probability estimates also differ, as might be expected, with respect to assumed differences in vulnerability relating to bridge foundation type, watercourse characteristics and the amount of resource committed to inspection and maintenance.

Expert judgements about fragility for any given bridge during a relatively modest flood event of 25-year return period indicated failure probabilities of around 1% or smaller, with uncertainties ranging from around 0.01% up to a few percent.

For an extreme flood with a 500-year return period, experts' central estimates suggest that a well-maintained bridge in a morphologically stable channel with modern or bedrock foundations has less than a 20% chance of failing due to scour, rising to nearer 50% for a poorly maintained bridge, or a bridge in an unstable channel on weak foundations; however uncertainty about these estimates is very wide, with experts judging that the true chance of failure could conceivably be less than 1% or nearly 95%.

Different assumptions about the foundations and watercourse type led to large variations in estimates of the uncertainty about failure probabilities under assumptions of no maintenance or routine (i.e. "business-as-usual") maintenance, particularly for the more extreme flood events (100-year and 500-year return periods).

Subjectively wide uncertainties were indicated in the group fragility estimates, reflecting a combination of differences in interpretation and, as revealed through calibration questions, differences between experts in their inherent assessments of uncertainties.

Increasing assumed levels of resourcing for monitoring and scour assessment translated into reductions in the experts' estimates of annual or flood-event failure probabilities, but these reductions were small relative to the experts' overall judgements of uncertainty, which were affected very little by those different assumptions. This finding appears to indicate some tension between qualitative statements, which stressed the importance of monitoring and assessment as a vital plank in scour risk management, "best" estimates of failure probabilities which reflect these statements to some extent, and judgements of uncertainty, which appear to remain very conservative under the three assumed levels of resourcing that we tested.

**6.3 Methodological findings**

The workshop format stimulated strong debate about the problem definition, and the different assumptions relevant in different countries, in particular relating to the age profile and physical scale of bridges and rivers when comparing, say, the UK with North America. As part of this process, the group had time during the workshop to debate and modify the elicitation questions, although the time available was necessarily constrained.

Some of the panel members commented during the workshop, and in subsequent feedback, that it would have been useful to define the context for each elicitation question in more detail. For instance, assumptions made about inspection and maintenance protocols may have led to differences in how individual experts interpreted those questions. If experts assumed that bridges are routinely inspected after any flood event, then the occurrence of sequences of events might be viewed as less important than other vulnerability factors because any problems found in the inspection would be addressed in a manner commensurate with the nature and extent of the problem. Under these circumstances, past flooding experience may not have been regarded as an important primary indicator of increased vulnerability. Feedback after the workshop indicated also that there could be differences of interpretation relating to the physical and engineering context for a particular structure. For example, the questions did not specifically distinguish between channels with cohesive versus non-cohesive sediments, or tidal versus non-tidal flows.

Following informal feedback and discussions with some of the group, we conclude that there would be merit in holding some form of initial consultation, prior to an elicitation workshop of this type, to establish whether an expert group feels the intended target questions are defined precisely enough, and with sufficient supporting contextual information to be interpreted

unambiguously. Bearing in mind that the aim of an elicitation is to gather evidence of experts' judgements about uncertainties, rather than their capacity to access information from the literature or other resources, there then would be a further challenge to provide sufficient but not excessive context material without inducing pre-judgement influences, such as availability bias.

When individual experts' estimates of failure probabilities were combined according to their uncertainty judgement weights,
validated against a set of control questions in the Classical Model analysis, the pooled uncertainty bounds became narrower relative to those produced by unweighted averaging, particularly for situations where a bridge is inherently resilient (i.e. lower failure probability cases); this appears to reflect a less tentative, less precautionary judgement about uncertainty for the most resilient asset types when compared with a naïve, uncritical appraisal of all experts' responses.

There are intangible benefits to be gained from fostering communication and discussion between internationally diverse groups
of experts from various different sectors, and the workshop, with its structured elicitation process, provided a constructive – and stimulating – forum for such exchanges.

## 6.4 Comparison with industry scour risk assessment guidance

In this study, the factors identified as important in assessment of scour risk are broadly consistent with industry guidance (summarised with a UK focus, but including reference to international good practice, by Kirby et al., 2015). Factors considered
by the expert group that do not have obvious counterparts within industry guidance, for either screening or detailed assessments, related to: sequences of events, expressed here in terms of the number of floods in recent years; construction date of a bridge; angle of the approach flow, and removal of weirs in the vicinity of a bridge (although the latter is considered in various contexts by Kirby et al., 2015 and Arneson at al., 2012). The expert group ranked none of the above factors within the nine most important factors.

This study was informed by a framework for risk analysis predicated on a probabilistic treatment of hazards and fragility, extending further than the "design event" concept adopted within most industry guidance. In UK scour management guidance, a detailed scour assessment involves estimating potential scour depth for a design event and comparing this with foundation depth. Starting from the perspective that failure probability is conditional on loading, which could be defined in many different ways, the study has explored formulations for a more general, probabilistic failure function and the associated uncertainties
about estimates of failure probabilities over a wide spectrum of load events. In assessing possible definitions of the load condition, the duration of a flood event and the possibility of sequences of events increasing the chance of a failure are regarded as important considerations, in addition to measures of peak hydraulic load. Flood return period, or exceedance probability, was considered as a standardised, probabilistic expression for the load condition in a fragility function.

Knowledge and data uncertainties are considered within industry guidance through a combination of qualitative and
quantitative measures. Here, a more explicit quantification of expert judgements about uncertainty was possible through the application of structured elicitation methods. Pooled judgements about uncertainty in scour failure probabilities are more tightly constrained by taking account of the empirical calibration of individual experts' accuracy in assessing uncertainties, although this effect diminished as more extreme, and therefore rarer, flood events were considered.

The experts' pooled estimates of failure probability reduced when considering scenarios involving increasing levels of
resources invested in scour assessment and maintenance. This appears to be consistent with the widespread use in practice of tiered risk management approaches involving generalised, high level screening followed by selective detailed assessments to enhance confidence in the mitigation of scour risk on a prioritised basis.

## 7. Conclusions

The elicitation workshop has provided to the authors' knowledge the first formal, pooled assessment of expert judgements
about scour risk uncertainties. It demonstrated that specialised elicitation methods, often previously applied for very extreme

natural and anthropogenic hazards, could be used successfully to investigate infrastructure failure risks that are subject to measurement and modelling uncertainties and are relatively infrequent, although not extremely rare compared with some other hazards. It has helped to provide a rational ordering of factors that could be considered in designing scour vulnerability assessment protocols and risk analysis models. The factors identified here were in line with international good practice in

industry, but also suggested that factors relating to hydraulic and morphological changes in watercourses, even some distance from a bridge, could be given more emphasis. A probabilistic measure of flood severity (flood flow return period) was ranked highly alongside physical variables (such as peak flow or flow velocity) when considered as a potential load variable in defining a fragility function.

The results of the study should not be read as substituting for modelled or empirically-derived estimates of scour vulnerability.

Rather, they add a view of broader uncertainties that are not easily captured in models or empirically-derived engineering formulae, and include uncertainties relating to subjective interpretations and judgements. In this sense the results help to reveal broad uncertainties about scour risk, and to highlight the continuing need for monitoring and research to constrain uncertainties about scour risk.

The heterogeneity of river environments, bridge types and engineering approaches found in different contexts makes it very

difficult to specify a generic scour fragility model. Despite these challenges, the group succeeded in reaching workable compromises about generic descriptions of bridges, maintenance regimes and risk factors that could be used, for the purposes set out in Sect. 2, in a quantitative fragility model.

After carefully debating the definition of terms, the group's input to a structured elicitation process enabled pooled estimates of scour fragility to be derived, expressed as the probability of a bridge failure conditional on flood events of varying severity,

where this severity was also expressed in probabilistic terms.  Although this study did not aim to develop a specific fragility model for immediate application, the results could help to guide and motivate the choice of loading variables in the development of scour fragility functions. By capturing experts' quantitative judgements about uncertainties in the assessment of failure probabilities, which were found to be wide, the results may provide additional context as part of an informed assessment of uncertainty within risk models developed in future.

The expert group repeatedly stressed the essential role of investment in scour assessment and maintenance.  Even so, the experts' weighted and pooled judgements about uncertainty remained wide regardless of whether assessment and maintenance was assumed to be more, or less, intensive than the *status quo*, suggesting that residual uncertainties remain, even after mitigation of the risk of scour, and that the residual risk of bridge failures remains important.

**Author contribution**

Rob Lamb prepared the manuscript with contributions from all co-authors, Willy Aspinall and Henry Odbert facilitated the elicitation exercise and carried out the data analysis.

**Acknowledgments**

We are grateful to Lisa Hill (University of Bristol) for project management support and to each of the participants in the expert group, listed below in alphabetical order, who gave their time to contribute so enthusiastically to the workshop:

Michael Beer (Professor of Uncertainty in Engineering, University of Liverpool); Jeremy Benn, (Executive Chairman, JBA Group): Kevin Dentith (Chief Engineer Bridges & Structures, Devon County Council); Rob Ettema (Professor of Civil and Architectural Engineering, University of Wyoming); Kevin Giles (Senior Project Engineer, Network Rail); Peggy Johnson (Professor of Civil Engineering, Penn State University); Andy Kosicki (Chief, Structure Hydrology and Hydraulics Division, MD State Highway Administration); John Lane (Structures Engineer, Rail Safety Standards Board); Caroline Lowe (Principal

Engineer, Network Rail); John McRobert (Highway Structures Unit, Department for Regional Development, Northern

Ireland); Bruce Melville (Professor of Civil and Environmental Engineering, University of Auckland); Chris Perkins (Senior Programme Manager, Asset Management, Network Rail); John Phillips (Environment Agency); Marta Roca-Collell (Principal Engineer, HR Wallingford); Max Sheppard (Principal, INTERA Incorporated); Thorsten Wagener (Professor of Civil Engineering, Bristol University); Bruce Walsh (Principal, Northwest Hydraulic Consultants); Lyle Zevenbergen (Hydraulic
5   Engineer, Tetra Tech Surface Water Group).

The elicitation was anonymised throughout and the analysis processed by a neutral facilitator. By the nature of the elicitation methodology, the findings presented here cannot be attributable to any individual. The subsequent interpretations presented in this paper are those of the authors.

This study was funded by the UK Natural Environment Research Council (NERC) under the Environmental Risks to
10  Infrastructure Innovation programme, grant number NE/M008746/1. Rob Lamb was funded by the JBA Trust (www.jbatrust.org, where further details of the workshop results can be found), project number W14-7290. Jim Hall (Oxford University) is thanked for proposing a bridge scour elicitation study following earlier work within the CREDIBLE consortium (NERC Grant NE/J017299/1), which also supported Willy Aspinall and Thorsten Wagener. We would like to thank the two anonymous referees and Prof. Bruno Merz for their constructive peer review and editorial suggestions.

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

**Table 1: Summary of specific questions posed in the elicitation workshop**

| Question | Motivation | Results |
|---|---|---|
| 1) What are the most important factors that should be considered in assessing scour risk to bridges? | | |
| What are the most important factors that should be considered in assessing scour risk to bridges? | To explore what variables could and should be chosen to describe the loading condition(s) relevant to scour risk assessment. | Section 5.1 Table 3 Figure 2 |
| What factors might be proposed to define relevant loading conditions for a scour fragility function? | | Section 5.1.1 Table 4 |
| What factors are important in determining how the risk of bridge failure may change? | To explore conditions that might provoke re-evaluation of scour risk, including the potential influence of climate change. | Section 5.1.2 Table 5 |
| 2) Quantitative elicitation of failure probabilities, with uncertainties | | |
| Elicitation of bridge failure probabilities, with uncertainty ranges, for specified flood events | To capture pooled expert judgements about scour failure probabilities (fragility), and the associated uncertainties, for bridges subjected to flooding. | Section 5.2.2 Figure 3 |
| Elicitation of annual failure probabilities | To explore the influence of implicit or explicit assumptions about flood event frequencies on expert judgements of uncertainty about bridge scour. | Section 5.2.3 Figure 4 |
| Elicitation of conditional event failure probabilities | To capture expert judgements about the scour failure probabilities, and associated uncertainties, for bridges subjected to a sequence of flood events. | Section 5.2.4 Figure 5 |
| Elicitation of triggers for asset inspection | To capture expert judgements about the severity (in terms of relative frequency) of a flood event that should trigger a precautionary bridge inspection. | Section 5.2.5 Table 6 |

**Table 2: Proposed vulnerability factors**

| Group | Proposed factors | Comments |
|---|---|---|
| Characteristics of the bridge structure | ▪ Foundation depth<br>▪ Foundation type<br>▪ Structure span<br>▪ Construction date<br>▪ Existence of scour protection<br>▪ Flow constriction at the bridge<br>▪ Bridge type | Relate to static characteristics of the structure. |
| Characteristics of the watercourse | ▪ Bed material<br>▪ Unstable watercourse | Factors relating to hydro-morphological situation in the river |
| Hydraulic conditions | ▪ Flow velocity<br>▪ Location on a river bend or confluence<br>▪ Oblique approach flow | Location on bend/confluence and oblique approach were included in view of their potential effects on velocity distributions and turbulence. |
| History and uncertainty about information | ▪ Application of scour assessment and monitoring procedures<br>▪ Whether there is a history of scour problems<br>▪ Whether or not foundation depth is known<br>▪ Whether or not foundation type is known<br>▪ Number of floods in the last 5 years<br>▪ History of debris accumulation | Broad group of factors reflecting how much is known about scour vulnerability at a bridge, including evidence from past events (especially previous occurrence of scour) and also whether the bridge characteristics are well known. |
| Change factors | ▪ Sand/gravel extraction in the reach near the bridge<br>▪ Weir has been removed near bridge | Changes at the bridge or elsewhere in the watercourse that could lead to changes in susceptibility to scour. |

**Table 3: Ranking scores for the importance of factors that should be considered in assessing scour risk to bridges (Question 1). Higher score indicates greater importance. "St. dev." is the score standard deviation derived from probabilistic inversion of experts' collective responses.**

| Item | Factor description | Piers | | Abutments | |
|---|---|---|---|---|---|
| | | Score | St. dev. | Score | St. dev. |
| 1 | Foundation depth | 0.61 | 0.26 | 0.59 | 0.28 |
| 2 | Foundation type | 0.63 | 0.32 | 0.53 | 0.28 |
| 3 | Whether foundation depth is known or not | 0.51 | 0.35 | 0.51 | 0.33 |
| 4 | Whether foundation type known is known or not | 0.43 | 0.32 | 0.43 | 0.29 |
| 5 | Bed material | 0.47 | 0.23 | 0.45 | 0.29 |
| 6 | Structure span | 0.25 | 0.24 | 0.39 | 0.31 |
| 7 | Scour history | 0.71 | 0.24 | 0.69 | 0.23 |
| 8 | Application of scour assessment and monitoring procedures (labelled "assmt/procedure") | 0.58 | 0.29 | 0.51 | 0.29 |
| 9 | Construction date | 0.33 | 0.16 | 0.30 | 0.24 |
| 10 | Flow velocity | 0.59 | 0.19 | 0.67 | 0.23 |
| 11 | Number of floods in the last 5 years | 0.32 | 0.23 | 0.39 | 0.27 |
| 12 | Existence of scour protection | 0.64 | 0.20 | 0.53 | 0.29 |
| 13 | Location on a river bend or confluence | 0.36 | 0.20 | 0.35 | 0.20 |
| 14 | Oblique approach flow | 0.48 | 0.26 | 0.34 | 0.24 |
| 15 | Constriction at bridge | 0.56 | 0.23 | 0.57 | 0.27 |
| 16 | Bridge type | 0.18 | 0.17 | 0.24 | 0.20 |
| 17 | History of debris accumulation | 0.57 | 0.26 | 0.50 | 0.26 |
| 18 | Unstable watercourse | 0.68 | 0.23 | 0.63 | 0.25 |
| 19 | Sand/gravel extraction in the reach near the bridge | 0.71 | 0.24 | 0.67 | 0.23 |
| 20 | Weir has been removed near bridge | 0.55 | 0.21 | 0.48 | 0.24 |

**Table 4: Ranking scores for factors according relevance to defining the loading condition for a scour fragility function**

| Item | Name | Score | St. dev. | Cluster |
|---|---|---|---|---|
| 9 | Frequency and amount of debris | 0.65 | 0.25 | A |
| 1 | Peak flow | 0.63 | 0.25 | A |
| 6 | Flow return period | 0.61 | 0.30 | A |
| 7 | Flow velocity relative to sediment critical flow | 0.59 | 0.26 | A |
| 3 | Time during which flow is greater than a critical threshold for scour initiation ("Time flow > threshold") | 0.59 | 0.26 | A |
| 2 | Peak water level | 0.45 | 0.26 | B |
| 4 | Time during which level is greater than a critical threshold for scour initiation ("Time level > threshold") | 0.45 | 0.26 | B |
| 5 | Number of "high flows" (capable of causing scour) in last year | 0.41 | 0.28 | B |
| 10 | Sediment concentration reaching the bridge at high flows ("High flow sediment concentration") | 0.34 | 0.25 | C |
| 8 | Application of scour assessment and monitoring procedures ("Assessment/procedure") | 0.31 | 0.23 | C |

**Table 5: Ranking scores for factors affecting change in scour vulnerability**

| Item | Name | Score | St. dev. |
|---|---|---|---|
| 4 | Inspection regime changes | 0.69 | 0.26 |
| 5 | Maintenance regime changes | 0.62 | 0.25 |
| 7 | Dredging up/downstream | 0.61 | 0.25 |
| 9 | Watercourse changes | 0.58 | 0.27 |
| 8 | Weir/dam removal | 0.54 | 0.25 |
| 6 | Flood defence construction | 0.52 | 0.24 |
| 2 | Catchment land manage changes | 0.47 | 0.27 |
| 1 | Climate change affects frequency of extreme rain | 0.22 | 0.20 |
| 3 | Bridge use demands | 0.22 | 0.19 |

**Table 6: Judgements about flood relative magnitude (in return period, years) appropriate to trigger asset inspection.**

| | Lower value (5th percentile) | Median (50th percentile) | Mean | Upper value (95th percentile) |
|---|---|---|---|---|
| Group estimate pooled with experts' weighted according to calibration questions | 1.0 | 5.6 | 15 | 48 |
| Group estimate pooled with experts' weighted equally | 1.2 | 26 | 94 | 318 |

| | Lower value (5th percentile) | Median (50th percentile) | Mean | Upper value (95th percentile) |
|---|---|---|---|---|

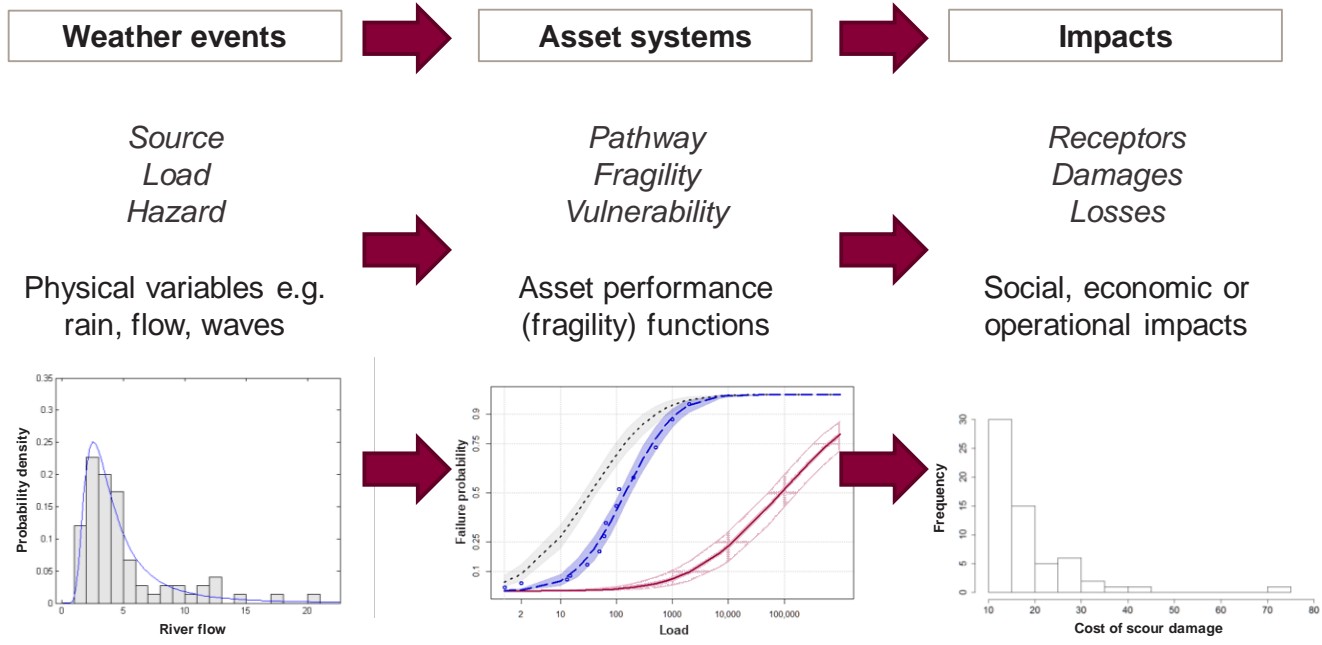

5    **Figure 1. High-level conceptual risk model**

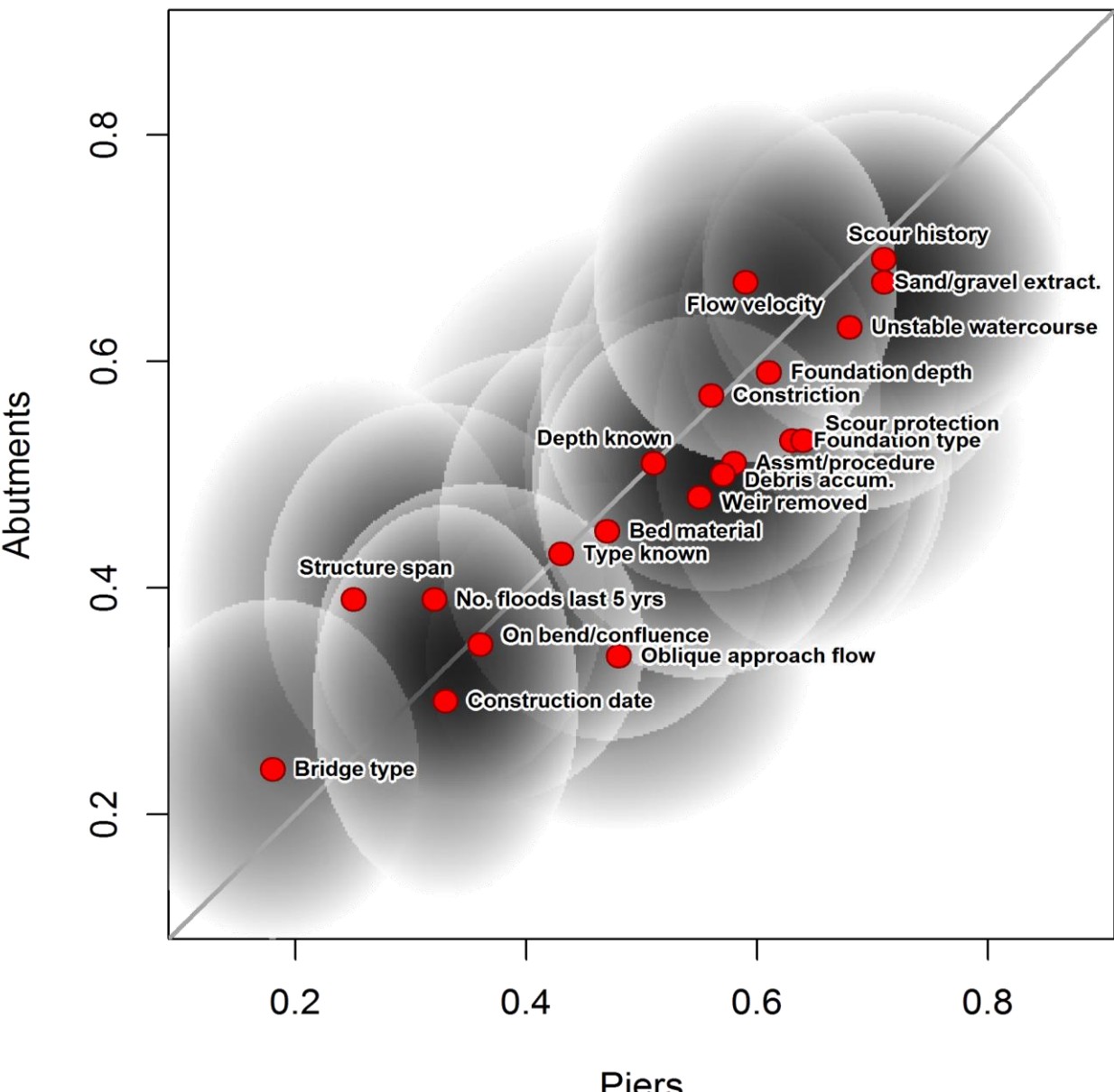

**Figure 2. Ranking scores (dimensionless) for experts' responses to Question 1: "What are the most important factors that should be considered in assessing scour risk to bridges?", comparing the responses when considering scour at bridge piers (horizontal axis) and at abutments (vertical axis). Higher scores indicate greater importance, in the judgements of the expert group. Ellipses show uncertainty about the scores, reflecting variation in the experts' responses to the question, and are 95 percentile contours of bivariate normal distributions around each score, with areas log-scaled by the geometric means of the associated standard deviations inferred from probabilistic inversion of experts' collective responses (Table 3). Ellipticity indicates differences in the pairs of standard deviations; larger areas for a factor indicate higher joint standard deviation about its score. A horizontally-extended ellipse indicates greater uncertainty about a factor's importance when considering its impact on scour at bridge piers compared with abutments; vertically extended ellipses indicate greater uncertainty about importance for scour at abutments.**

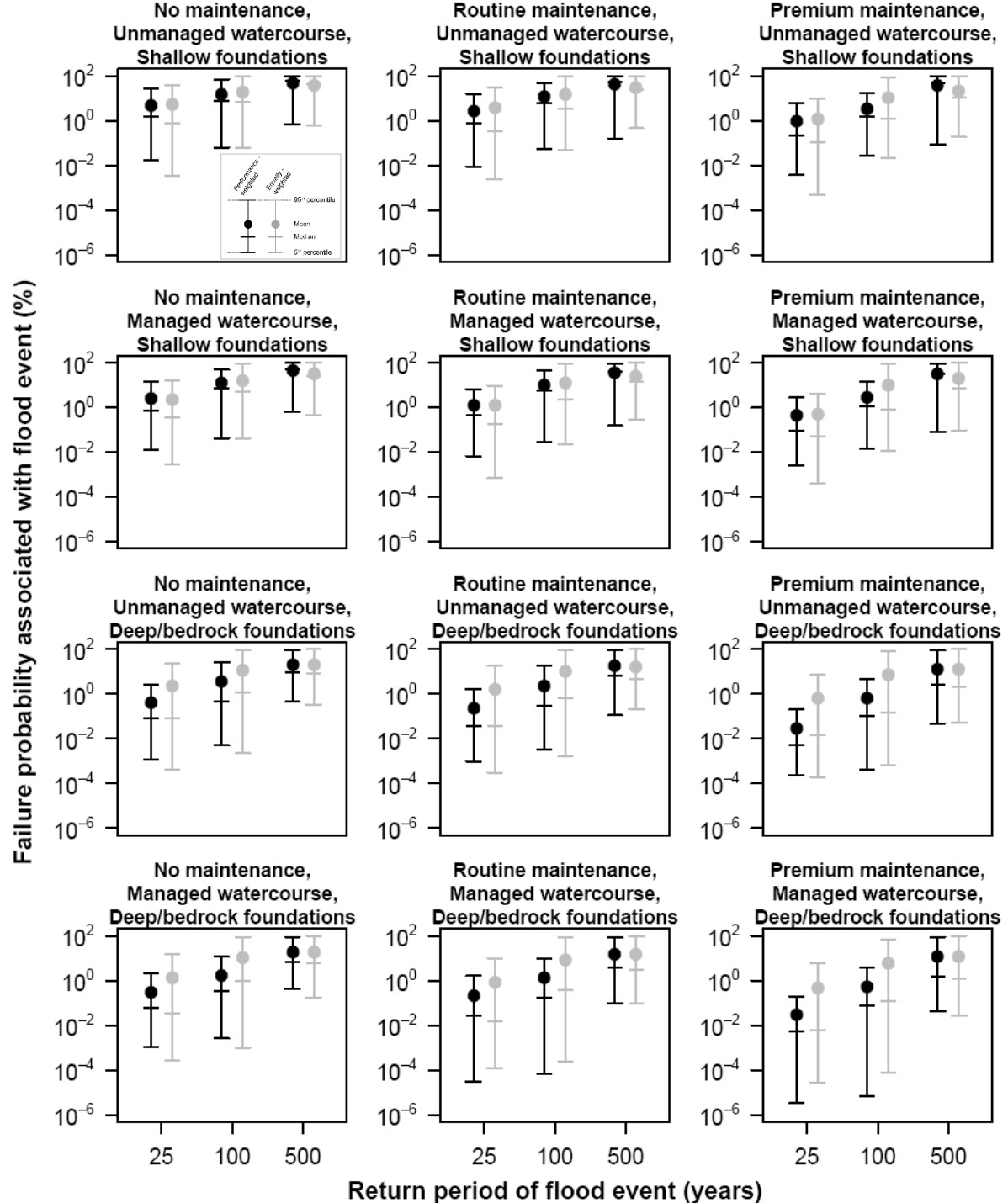

**Figure 3. Fragility estimates for bridge failure probability as a function of flood event severity, expressed in terms of the return period of the flood event. Solid lines represent performance-weighted pooled expert judgements; light grey lines are unweighted pooled expert judgements. Whiskers indicate the 5th and 95th percentile uncertainty ranges around the mean (filled circle) and median (horizontal bar) expert estimates.**

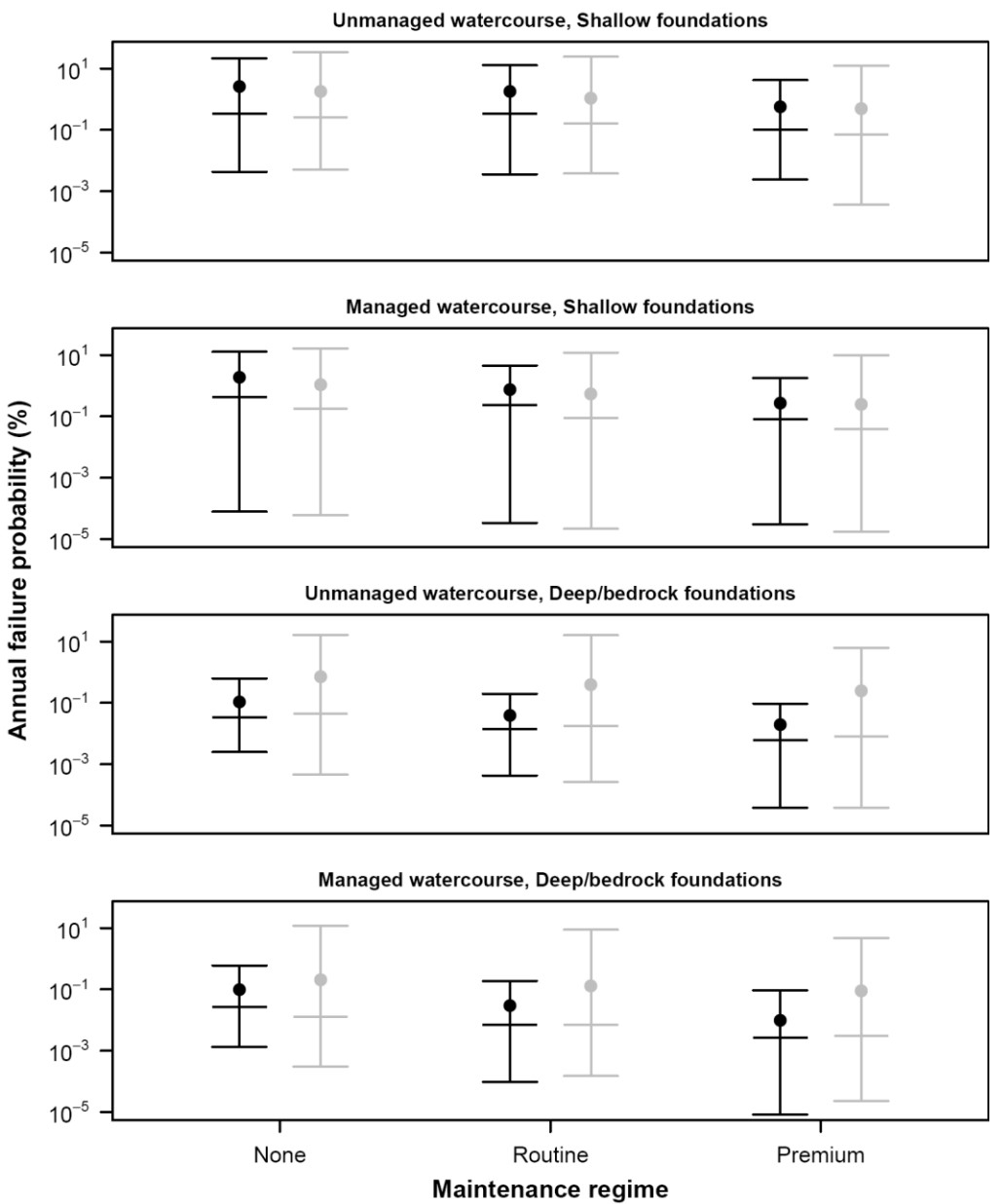

**Figure 4. Fragility estimates for annual unconditional bridge failure probability under three assumed monitoring and maintenance ("maintenance") regimes.**

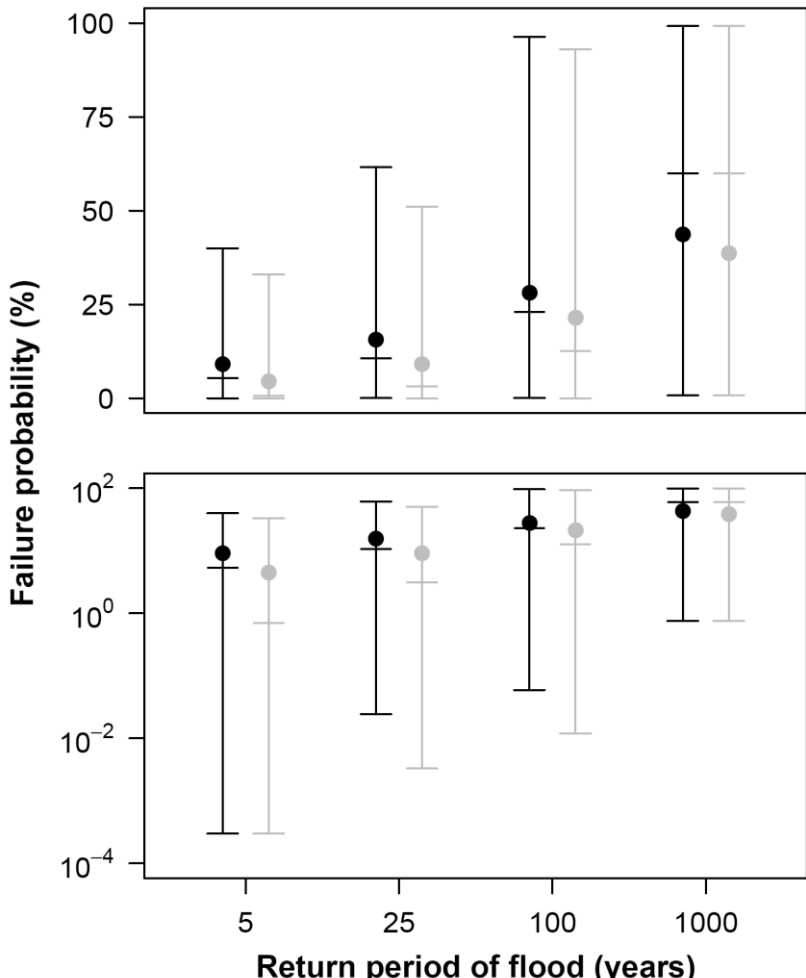

**Figure 5. Estimated bridge failure probabilities as a function of flood event severity, expressed in terms of the return period of the flood event, conditional on a preceding flood event of 100-year return period having occurred with no intervening maintenance action. Upper and lower panel show the same data, plotted on different scales.**