# Peer review of "Vulnerability of bridges to scour: insights from an international expert elicitation workshop"

_Natural Hazards and Earth System Sciences, 2016_

## Referee Comment (RC1) · Anonymous Referee #1 · 20 Dec 2016

GENERAL COMMENTS

The manuscript prepared by Rob Lamb, Willy Aspinall, Henry Odbert and Thorsten Wagener, provides the context and the outcomes of an elicitation workshop held in London in 2015 with international experts on bridge scour risk assessment. Scour of bridge piers or abutments during severe flood events may lead to infrastructure failure, from lower safety levels to dramatic collapse. Inherently uncertain, scour risk is difficult to quantify from available data, especially for risk analyses where a network of numerous assets is considered. The authors of the manuscript claim that expert knowledge and judgement represent a valuable source of data. With this expert elicitation process, they intend to lay the ground for the development of fragility functions that may be applied for broad scale network risk analysis. During the workshop, the experts were asked to (1) rank the vulnerability factors that should be considered in assessing risk

of scour, and (2) quantify the failure probabilities of bridge due to scour under a number of loading conditions, with associated uncertainties. The answers were analysed based on a variant of the survey method of paired comparison for (1), and a structured expert judgement procedure for (2). To the authors' knowledge, the formal process of elicitation undertaken is unique in the field of scour risk. The results of the workshop are scrutinised, compared to industry guidance, and the methodology evaluated.

This manuscript provides an interesting approach to inform risk of scour, made possible thanks to a collective endeavour of international experts, and appear to be based on a sound methodology. Overall, it is well written and the results are nicely detailed, nevertheless, minor revisions are necessary to make a few points clearer.

SPECIFIC COMMENTS

The study presented by the authors is based on a formal process of elicitation whose techniques are described in the section 4 'The role of expert elicitation'. However, they are too briefly described and do not allow the reader to fully understand the following section 5 where the results are analysed. In particular, the manuscript would benefit from a more detailed presentation of the Classical Model used to tackle the second question raised in the study. In the section 1, the methods applied to weight information in the group of experts are succinctly mentioned. As these methods were used throughout all the process of elicitation and appear in most of the results and figures, they should be mentioned in the section 4 and further detailed.

In the section 3, it is stated that the first question investigated in the study was "What variables should be chosen to describe the loading conditions relevant to scour risk?". In the section 5.1 'Question (1): Vulnerability factors that should be considered in assessing risk of scour', the first question asked to the expert appeared to be "What are the most important factors that should be considered in assessing scour risk to bridges?". These three different expressions of what was the first question addressed in the study are confusing for the reader. Using the same terminology and defining

which from either the loading conditions and/or vulnerability factors were screened should help to make the aim of the study clearer.

The details of the questions asked to the experts are actually all presented in the section 5, which is the section of the results. This section is thus easy to read, each question is stated in the relevant sub-section and the results directly analysed. However, it makes the methodology and its overall objectives more difficult to understand for the reader. For instance, the question asked and presented in the section 5.2.5 about the triggers for asset inspection almost comes as a surprise, which should not be the case there. Thus, all the questions asked to the experts could be listed in one of the first sections, and their objectives made clearer.

The three above specific comments raise issues regarding the methodological presentation of the study. The manuscript could be improved in providing a clearer, more structured and outlined, description of the methodology. It would help to highlight the fact that the process of elicitation undertaken by the group of international experts is formal and objective, which is a strength of the study. Potentially, the section 4 could be renamed 'Methodology' and adapted accordingly.

The number of experts who contributed to the workshop is not provided in the manuscript. It could be relevant as statistical methods are applied to infer global results from their answers.

It is highly appreciated that the authors mentioned and reported the discussions that took place during and after the elicitation process.

As written in the first few lines of the manuscript, with this study the ultimate goal of the authors is to "inform the development of fragility functions that may be applied within a broad scale risk modelling framework". From the conclusion, it is not clear how in practice the results from the elicitation workshop could be used in order to achieve this goal, or what future work would be required.

[Figure]

TECHNICAL CORRECTIONS

Page 3, line 28: Replace "sour" with "scour"

Page 5, line 9: Aren't there any references available for the hazard-vulnerability-loss concepts?

Page 6, line 2: Replace "form" with "from"

Page 6, lines 7-8: Replace "(see, for example Decò and Frangopol 2011)" with "(see, for example, Decò and Frangopol, 2011)

From page 8, line 29, to page 9, line 2: In this paragraph, the intensity indicators flood flow, flood velocity and flood return period are compared, but, the outcome of this comparison is difficult to understand. Besides, I'm curious about the ability of the eliciting method used in the study to deal with dependent or "intrinsically linked" factors, such as these three factors.

Page 10, line 3: Replace "to have to in mind" with "to have to bear in mind"?

Page 11, line 13: Aren't the lower uncertainty bounds that vary largely?

Page 12, paragraph from line 4 to line 10: The last sentence of this paragraph is hard to understand.

Page 12, line 28: "flood rarity" could be replaced with "flood severity" for more consistent use of the terminology. If so, it could also be replaced in the title of the Figures 3 and 5.

From page 12, line 25, to page 13, line 4: Flood frequencies are expressed under the forms of probability, 1-in-XX AEP, return period and 1/XX AEP. Although they are all equivalent, for the reader it would be more comfortable to get all the flood frequencies detailed in this paragraph under the same form, such as the return period as shown in Table 5.

Author contribution and Acknowledgements: The full name of the authors should be written instead of their initials.

Figure 2: The font size of the labels is rather small and should be increased. Although the ellipses represent a nice graphical way to represent the 95% confidence bounds of the elicited parameters, I'm afraid that there are too many parameters plotted. The ellipses of only a few of them are visible, and not so clearly.

Figure 3: The graphical legend in the top left frame is definitely too small. Instead, the description of the values represented in the figure should be incorporated directly as text in the legend.

[Figure]

---

## Referee Comment (RC2) · Anonymous Referee #2 · 11 Mar 2017

The paper was written by R. Lamb et al. "Vulnerability of bridges to scour: insights from an international expert elicitation workshop" reports the results of the workshop held on bridge scour vulnerability. A topic is very interesting and common in the engineering world but loaded with a number of uncertainties in the influenced factors and estimation methods. The paper reports about the workshop outcomes and their statistical evaluation. The paper could be divided into two parts: results of the workshop and statistic evaluation of the expert's opinions. In the first part, the experts detect and define the factors influencing the bridge scour and also give their (expert) judgment of the importance of each factor. The conclusions of this part of the paper could be seen as "state of art" view on the scour influencing factors. Also, the experts ranked the factors and gave their opinion about the importance of each factor. The detected factors have then been statistically evaluated and each factor is loaded with a mean value and standard

deviation. In the next part, the authors applied statistical and probability methods and evaluate statistically all important parameters. The paper does not give an impression of the number and academic structure of the expert group. Therefore the factors, but especially the statistical evaluations could be seen also as the subjective opinion of a group of persons. Based on the clarifications in the text it could be concluded that the most of the experts originate from UK and USA. The intention of the paper is just to detect the factors and influences, without any vision and consideration on the mathematical evaluation, modeling of the score vulnerability or methods for the bridge scour risk reduction. Therefore the publication could be used as a tool for detection if the scour influencing factors, but not giving any answer on the mentioned scour mitigation measures as well as definition of the maintenance level, specified as none, routine or premium. The mathematical evaluation of the fragility estimates presented in the Figures 3-5 is difficult to be followed and in some cases gives misleading or less explicit answers, especially in the case of maintenance (Figure 3). The comparison of figure 3 within 3*4 diagrams are presented is difficult for comparison and distinguishing. Discussion and Conclusion chapter is too extensive and therefore unclear, striving to an additional summary that will really summarize the findings of the work.
* * *

---

## Author Comment (AC1) · 14 Jun 2017

nhess-2016-350

**Vulnerability of bridges to scour: insights from an international expert elicitation workshop**

**Rob Lamb, Willy Aspinall, Henry Odbert, and Thorsten Wagener**

**Response to Referee #1**

**GENERAL COMMENTS**

*REFEREE COMMENT*

The study presented by the authors is based on a formal process of elicitation whose techniques are described in the section 4 'The role of expert elicitation'. However, they are too briefly described and do not allow the reader to fully understand the following section 5 where the results are analysed. In particular, the manuscript would benefit from a more detailed presentation of the Classical Model used to tackle the second question raised in the study. In the section 1, the methods applied to weight information in the group of experts are succinctly mentioned. As these methods were used throughout all the process of elicitation and appear in most of the results and figures, they should be mentioned in the section 4 and further detailed.

*RESPONSE*

We will add the following text to provide further explanation of the Classical Model:

For uncertainty quantification, a structured expert judgment procedure formulated by Cooke (1991), known as the "Classical Model", was adopted in this study. This approach is supported by a software package called EXCALIBUR (Cooke and Solomatine, 1992), available at www.lighttwist.net/wp/excalibur. This is a quantitative elicitation method used to assess numerical estimates of uncertain parameters or variables, in this case scour failure probabilities conditional on various stated assumptions.

The unique feature of this approach is that distinct weights are given to individual experts, based on a statistical test of the expert's ability to judge uncertainties, determined empirically by performance metrics derived from control questions. The main steps in the procedure for applying the Classical Model in practice are:

- A group of experts is selected by a problem owner and a facilitator, and an elicitation protocol is developed; this comprises a set of multiple 'seed items' (i.e. the control) and a set of 'target questions', both drawn from within the experts' field of knowledge;
- The experts assess the set of 'seed item' quantities; experts are not expected to know the true values but should be able to capture most of them by defining informative credible ranges. Taking their responses to the set of seed items, the experts are treated as statistical hypotheses and are scored with respect to statistical likelihood ('calibration') and informativeness, using theory and procedures described by Cooke (1991);
- These scores are combined to form individual performance weights using scoring rules formulated such that experts receive maximal weight by, and only by, stating their true degrees of belief;
- The elicitation protocol includes a set of 'target item' questions; in principle, these could be subject to possible measurement or observation but, in the problem owner's case, for one

reason or another they are not amenable to such an approach; the only feasible recourse is to seek expert judgements;

- Experts are elicited individually regarding their uncertainty judgements for these target items. A weighted linear combination of their responses is calculated for each question using EXCALIBUR to provide a pooled result (known as a synthetic 'decision maker'), conditioned on the performance-weighted scores.

The latter is the key feature of this method. When it comes to attempting to resolve differences in expert judgments, searching for harmony of views by negotiation or conciliation can leave participants discomfited by the outcomes. Extensive experience (see below for references to previous case studies) overwhelmingly confirms that experts grow to favour the Classical Model approach because its performance measures are objective and amenable to diagnostic examination. The 'reward' nature of weights is very important. An expert's influence on the pooled result should not appear haphazard, and he/she should be discouraged from attempting to game the system by attempting to tilt his/her assessments to achieve a desired outcome. Thus, it is necessary to impose a formal scoring rule constraint on the weighing scheme. This means an expert achieves maximal expected weight by, and only by, stating assessments in conformity with their true scientific or technical beliefs.

In the section 3, it is stated that the first question investigated in the study was "What variables should be chosen to describe the loading conditions relevant to scour risk?". In the section 5.1 'Question (1): Vulnerability factors that should be considered in assessing risk of scour', the first question asked to the expert appeared to be "What are the most important factors that should be considered in assessing scour risk to bridges?". These three different expressions of what was the first question addressed in the study are confusing for the reader. Using the same terminology and defining which from either the loading conditions and/or vulnerability factors were screened should help to make the aim of the study clearer.

We agree, and have revised the terminology used in setting out Question 1 to be consistent.

The details of the questions asked to the experts are actually all presented in the section 5, which is the section of the results. This section is thus easy to read, each question is stated in the relevant sub-section and the results directly analysed. However, it makes the methodology and its overall objectives more difficult to understand for the reader. For instance, the question asked and presented in the section 5.2.5 about the triggers for asset inspection almost comes as a surprise, which should not be the case there. Thus, all the questions asked to the experts could be listed in one of the first sections, and their objectives made clearer.

We have added a table at the end of Section 4 summarising the questions posed to the expert group, their motivation, and where the results are presented and discussed.

**Table 1: Summary of questions posed in the elicitation workshop**

| Question | Motivation | Results |
|---|---|---|
| 1) What are the most important factors that should be considered in assessing scour risk to bridges? | | |

| What are the most important factors that should be considered in assessing scour risk to bridges? | To explore what variables could and should be chosen to describe the loading condition(s) relevant to scour risk assessment. | Section 5.1
Table 3
Figure 2 |
|---|---|---|
| What factors might be proposed to define relevant loading conditions for a scour fragility function? | | Section 5.1.1
Table 4 |
| What factors are important in determining how the risk of bridge failure may change? | To explore conditions that might provoke re-evaluation of scour risk, including the potential influence of climate change. | Section 5.1.2
Table 5 |
| 2) Quantitative elicitation of failure probabilities, with uncertainties | | |
| Elicitation of bridge failure probabilities, with uncertainty ranges, for specified flood events | To capture pooled expert judgements about scour failure probabilities (fragility), and the associated uncertainties, for bridges subjected to flooding. | Section 5.2.2
Figure 3 |
| Elicitation of annual failure probabilities | To explore the influence of implicit or explicit assumptions about flood event frequencies on expert judgements of uncertainty about bridge scour. | Section 5.2.3
Figure 4 |
| Elicitation of conditional event failure probabilities | To capture expert judgements about the scour failure probabilities, and associated uncertainties, for bridges subjected to a sequence of flood events. | Section 5.2.4
Figure 5 |
| Elicitation of triggers for asset inspection | To capture expert judgements about the severity (in terms of relative frequency) of a flood event that should trigger a precautionary bridge inspection. | Section 5.2.5
Table 6 |

The manuscript could be improved in providing a clearer, more structured and outlined, description of the methodology.

We hope that the fuller description of the Classical Model (see above) provides the required clarity within the existing structure setting out:

(A) the combination of two elicitation approaches (paired comparison implemented with the UNIBALANCE method, followed by the Classical Model pooled elicitation of uncertainties), and,

(B) the steps taken to implement the Classical Model method.

It would help to highlight the fact that the process of elicitation undertaken by the group of international experts is formal and objective, which is a strength of the study.

We thank the reviewer for suggesting further emphasis of this point, which we agree is an important feature of the study.

We previously stated in Section 4 that the method we adopted is "formalised" and "designed to tie results into stated and transparent methodological rules, with the goal of treating expert judgements in the same way as other scientific data in a formal decision process".

We have further emphasised in the new methodology text that experts have favoured the Classical Model because its "performance measures are objective and amenable to diagnostic examination", and discussed how those performance measures are derived from a set of control (seed) questions.

Potentially, the section 4 could be renamed 'Methodology' and adapted accordingly.

Section 4 has been re-titled "Expert elicitation methodology".

The number of experts who contributed to the workshop is not provided in the manuscript. It could be relevant as statistical methods are applied to infer global results from their answers.

We stated the number of experts on line 2 of the main text, along with their nationalities and the sectors they represented. We have added further detail to this description and corrected a typo error in the original number count.

It is highly appreciated that the authors mentioned and reported the discussions that took place during and after the elicitation process. As written in the first few lines of the manuscript, with this study the ultimate goal of the authors is to "inform the development of fragility functions that may be applied within a broad scale risk modelling framework". From the conclusion, it is not clear how in practice the results from the elicitation workshop could be used in order to achieve this goal, or what future work would be required.

It was not our aim to develop a new fragility model or protocol for industry application. However, we believe that the present study could help to guide and motivate the choice of loading variables and the structure of fragility functions. Furthermore, by capturing experts' judgements about (very uncertain) failure probabilities, we have created an evidence base that may be compared with such functions in future as part of an informed assessment of uncertainty.

We have added text to this effect in the conclusions (Section 6).

From page 8, line 29, to page 9, line 2: In this paragraph, the intensity indicators flood flow, flood velocity and flood return period are compared, but, the outcome of this comparison is difficult to understand. Besides, I'm curious about the ability of the eliciting method used in the study to deal with dependent or "intrinsically linked" factors, such as these three factors.

We have amended the text slightly with the aim of clarifying that the flood return period is being interpreted as a probabilistic definition of the load event.

The physical correlation in question relates to the inference of factor rankings, and we note that the three variables (flow, velocity and return period) were ranked 2nd, 3rd and 4th in the group's pooled scoring, which appears to be consistent with them being recognised by the experts as physically correlated.

**TECHNICAL CORRECTIONS**

Page 3, line 28: Replace "sour" with "scour"

Corrected

Page 5, line 9: Aren't there any references available for the hazard-vulnerability-loss concepts?

Added reference to K Mitchell-Wallace, M Jones, J Hillier & M Foote (eds), 2017, Natural Catastrophe Risk Management and Modelling: A Practitioner's Guide. Wiley-Blackwell, 536 pp. 218-229.

Page 6, line 2: Replace "form" with "from"

Corrected

Page 6, lines 7-8: Replace "(see, for example Decò and Frangopol 2011)" with "(see, for example, Decò and Frangopol, 2011)

Corrected

From page 8, line 29, to page 9, line 2: In this paragraph, the intensity indicators flood flow, flood velocity and flood return period are compared, but, the outcome of this comparison is difficult to understand. Besides, I'm curious about the ability of the eliciting method used in the study to deal with dependent or "intrinsically linked" factors, such as these three factors.

See response given earlier.

Page 10, line 3: Replace "to have to in mind" with "to have to bear in mind"?

The intended wording was "to have in mind". Corrected.

Page 11, line 13: Aren't the lower uncertainty bounds that vary largely?

Both upper and lower bounds vary. However, the differences in upper bounds are larger (bearing in mind the log scale of the probability axis). We have added a note to the text to highlight comparison of top left and bottom left panels in Fig. 3, noting the logarithmic scale.

Page 12, paragraph from line 4 to line 10: The last sentence of this paragraph is hard to understand.

We have revised this text as follows to clarify the point being made:

A discussion was held about whether the annual failure probability is in fact determined completely by design standards (i.e. the as-built performance of the bridge matches the desired design standard perfectly), effectively removing uncertainty about bridge vulnerability. This view would appear to

imply a standard of asset maintenance and that may be unachievable in practice and seems to be counter to the wide uncertainties about vulnerability to scour that emerged from the expert group elicitation. Empirically, historical evidence from the UK railway network shows that bridge failures have occurred under a wide range of flood conditions (van Leuwen and Lamb, 2014), suggesting that it is not appropriate to treat vulnerability deterministically.

Page 12, line 28: "flood rarity" could be replaced with "flood severity" for more consistent use of the terminology. If so, it could also be replaced in the title of the Figures 3 and 5.

We thank the reviewer for pointing this inconsistency out. We have changed the main text to read "…an inspection trigger based on a probabilistic measure of flood severity…" and revised the figure captions as suggested.

From page 12, line 25, to page 13, line 4: Flood frequencies are expressed under the forms of probability, 1-in-XX AEP, return period and 1/XX AEP. Although they are all equivalent, for the reader it would be more comfortable to get all the flood frequencies detailed in this paragraph under the same form, such as the return period as shown in Table 5.

We have made the suggested changes.

Author contribution and Acknowledgements: The full name of the authors should be written instead of their initials.

We have made the suggested changes.

Figure 2: The font size of the labels is rather small and should be increased. Although the ellipses represent a nice graphical way to represent the 95% confidence bounds of the elicited parameters, I'm afraid that there are too many parameters plotted. The ellipses of only a few of them are visible, and not so clearly.

We have revised the plot to increase the font size of the labels.

We agree that the ellipses obscure each other. In fact, this is an important feature of the results, which merely show the situation as it is:  there is a wide uncertainty enveloping many of factors. If the relative importance of the factors were known without uncertainty then there would not be a need to call upon expert judgment. Hence the overlap between ellipses reflects the motivation for the study.

Despite this, the factors at each end of the ranking scale are visibly separated.  The message the plot conveys from the elicitation is that the expert group produced a collective ranking order from a range of disparate individual views, with diversity of those views being captured by the ellipses.

The numerical values underlying the plot are reported in Table 3. We have added a note to say this in the figure caption.

Figure 3: The graphical legend in the top left frame is definitely too small. Instead, the description of the values represented in the figure should be incorporated directly as text in the legend.

We have made the suggested change.

---

## Author Comment (AC2) · 14 Jun 2017

**Vulnerability of bridges to scour: insights from an international expert elicitation workshop**

Rob Lamb, Willy Aspinall, Henry Odbert, and Thorsten Wagener

**Response to Referee #2**

**GENERAL COMMENTS**

*REFEREE COMMENT*

The paper could be divided into two parts: results of the workshop and statistic evaluation of the expert's opinions.

*RESPONSE*

The statistical evaluation was based directly on the experts' stated judgements using the structured elicitation methods and question protocols (Section 4) that were introduced to and accepted by expert group. The authors did not select or modify the statistical analysis methods ex-post, hence these statistical results represent "results of the workshop".

In the next part, the authors applied statistical and probability methods and evaluate statistically all important parameters.

We selected a sub set of the important factors, guided by group discussion, and explored in more detail the experts' judgements about bridge failure probabilities conditional on those factors. We did not evaluate all potentially important parameters in the elicitation of failure probabilities and uncertainties for reasons of time constraints and prioritisation of effort.

We have expanded the description of the statistical methodologies used to clarify their role in (a) importance ranking of potential causative factors, and, (b) elicitation of conditional failure probabilities (as set out in more detail in the response to Referee #1).

The paper does not give an impression of the number and academic structure of the expert group.

We have included a breakdown of participant' territories and sectors in the introductory paragraph.

Therefore the factors, but especially the statistical evaluations could be seen also as the subjective opinion of a group of persons.

The elicitation process necessarily relies on the subjective (but informed) judgements of the experts, which is a feature of estimation under conditions of profound uncertainty.

However, as discussed in Section 4, the elicitation methodology itself is designed to enable those subjective judgements to be collected, summarised and assessed in as transparent and objective a form as possible – this is the underlying motivation for the Classical Model as is discussed in the paper and well documented in the cited references.

The intention of the paper is just to detect the factors and influences, without any vision and consideration on the mathematical evaluation, modeling of the score vulnerability or methods for the bridge scour risk reduction.

The intention is, as stated, to support future development of fragility functions, and hence generic scour scale risk assessment models, by gathering systematically expert views on the identification of factors that may be considered in characterising a fragility function loading condition, and in formalising the judgement of a panel of experts about the magnitude of uncertainty around such functions.

We recognise that the technical methods for the elicitation were introduced only in summary form, and have added further description in Section 4 to provide a more detailed account. In the interests of brevity and of maintaining focus on the substance of the workshop, we have not presented the mathematical basis for the elicitation methods. However, the mathematical formulation is readily discoverable in the references cited within the text and we have added explicit pointers to those references.

The study relates to assessment of scour vulnerability, and the management of uncertainty in that assessment, therefore detailed discussion or evaluation of methods for bridge scour reduction is not within the scope of the paper (other than in the sense that improved knowledge of scour vulnerability may help to achieve this general goal).

Therefore the publication could be used as a tool for detection if the scour influencing factors, but not giving any answer on the mentioned scour mitigation measures as well as definition of the maintenance level, specified as none, routine or premium.

We have added further discussion in Section 6 about how the results may support the development of fragility functions and hence risk assessment models. The aim was not to determine specific mitigation measures or identify specific strategies relating to the three generalised maintenance levels.

The mathematical evaluation of the fragility estimates presented in the Figures 3-5 is difficult to be followed and in some cases gives misleading or less explicit answers, especially in the case of maintenance (Figure 3). The comparison of figure 3 within 3*4 diagrams are presented is difficult for comparison and distinguishing.

As mentioned above, we have added further detail about the evaluation of the fragility estimates, including explicit references to sources in which the mathematical analysis is described in full.

We are not sure why the estimates presented in Figs 3-5 are considered "misleading or less explicit". The reviewer does not offer a frame of reference against which our results are claimed to be "less explicit", and it is therefore difficult to respond to this point. The graphs and tables in the paper are empirical results of the elicitation process, which we believe are described clearly and with appropriate reference to background literature.

We are not sure what the reviewer means by "The comparison of figure 3 within 3*4 diagrams are presented is difficult for comparison and distinguishing", but are confident that the Figure and

accompanying text provide a clear summary of the results which can be interpreted with reference to terms defined within the paper. The results are complex, but this is an inevitable consequence of presenting a detailed view of the analysis and we contend that the salient features of the figures are identified and discussed within the text in Sections 5 and 6.

Discussion and Conclusion chapter is too extensive and therefore unclear, striving to an additional summary that will really summarize the findings of the work.

The Dicsussion and Conclusions section comprises four clearly labelled sub-sections, each with between 4 and 6 paragraphs of text. We do not agree that this is too extensive and would not necessarily equate the length of the section with its clarity.

We have observed that this comment does not align with the view of Reviewer #1, who stated that "Overall, it is well written and the results are nicely detailed".

---

## Author Comment (AC3) · 14 Jun 2017

Revisions marked as tracked changes.

Please also note the supplement to this comment:
http://www.nat-hazards-earth-syst-sci-discuss.net/nhess-2016-350/nhess-2016-350-AC3-supplement.pdf
* * *

---

## Author Comment (AC4) · 15 Jun 2017

We are grateful to the referees and editor for their comments and guidance, which have helped us revise the manuscript.

The supplement is a further revision, addressing comments raised by Referee #2 and the editor, in which a short Conclusions section has been added.

Please also note the supplement to this comment:
http://www.nat-hazards-earth-syst-sci-discuss.net/nhess-2016-350/nhess-2016-350-AC4-supplement.pdf
* * *
[Figure]

2016-350, 2016.

**Supplement:**

[revised manuscript text omitted]